# Integrated Analysis of Transcriptomes and Pectinase Gene Families Reveals a Novel Pathway Mediating Tomato Fruit Malformation

**DOI:** 10.3390/ijms262110739

**Published:** 2025-11-04

**Authors:** Junqin Wen, Quanhui Li, Xiaoyan Tao, Rong Zhou, Chaofan Yan, Qiwen Zhong

**Affiliations:** 1Academy of Agriculture and Forestry Sciences, Qinghai University, Xining 810016, China; liquanhui_2008@163.com (Q.L.); 18909710107@163.com (X.T.); yanchaofan00@163.com (C.Y.); 2Key Laboratory for Protection and Genetic Improvement of Qinghai Tibet Plateau Germplasm Resources, Qinghai University, Xining 810016, China; 3Sanya Research Institute, Nanjing Agricultural University, Nanjing 210095, China; zhour@njau.edu.cn

**Keywords:** transcriptomic analysis, malformed fruit, pectinase gene, tomato

## Abstract

Tomato fruit malformation causes substantial yield and economic losses, but its molecular mechanisms are not well understood. This study compared floral traits of WT ‘QT57’ and malformed-fruit mutant ‘QT2’, integrated transcriptomic data, and qRT-PCR analysis to screen key candidate genes, and analyzed the pectinase gene family. The results found the ‘QT2’ mutant differed from WT ‘QT57’ in flower and fruit development. Expression analysis of CLAVATA-WUSCHEL pathway genes preliminarily validated the compensatory mechanism of *SlCRCa* and *SlCRCb* in ‘QT2’ malformed fruit. Six pectinase genes were identified as key candidates via RNA-seq and qRT-PCR analysis. Transcriptomic and qRT-PCR analyses of the pectinase gene family revealed their potential role in regulating tomato fruit malformation. Family analysis showed 34 pectinase genes distributed unevenly across 12 chromosomes. Subcellular localization confirmed SlPL7 in the nucleus and SlPME9 in the cell membrane/endoplasmic reticulum. The PL and PME genes were evolutionarily close, suggesting a potential functional overlap. Gibberellin-responsive elements were found in most pectinase genes. Pectinase genes may regulate tomato fruit malformation through the gibberellin-WUS pathway, carbohydrate metabolism, or cell wall metabolic disorder. This pathway provides new targets gene for the precise regulation of fruit malformation and offers significant reference value for practical production.

## 1. Introduction

Tomato (*Solanum lycopersicum* L.) is a globally significant vegetable crop, prized for its unique fruit flavor, abundant nutritional content, and diverse culinary applications [1]. The occurrence of malformed fruits—deformities developing at the blossom end—often results in substantial market value losses for tomatoes [2]. Surveys conducted in regions such as Shanghai, Jilin, and Liaoning in China revealed that the rate of malformed tomatoes usually exceeded 20%, with rates reaching as high as 70% to 80% in protected cultivation areas during winter and spring, severely impacting tomato yield and quality [3]. Malformed tomato fruit has become a major challenge in tomato production.

The causes and molecular mechanisms of abnormal fruits have been studied. Flowers are generated from stem cells located in floral meristems (FM), and once the initiation of floral organs has occurred, the activity of the stem cells is arrested, and the FM is determined to form the floral organs. These events were defined as floral determinacy [4]. However, once the timing of FM termination is repressed, resulting in malformed fruits characterized by additional carpels and extra locules [5]. The mechanisms involved in this phenomenon can be regulated by CLAVATA-WUSCHEL feedback loops. The homeodomain transcription factor WUSCHEL (WUS) plays a key role in sustaining stem cell activity within the FM, as mutations leading to flowers with additional carpels result in malformed fruits [6]. Once carpel primordia are initiated, the tomato CRC paralogues (SlCRCa and SlCRCb), INHIBITOR OF MERISTEM ACTIVITY (SlIMA), and (KNUCKLES) SlKNU interact with HISTONE DEACETYLASE1 (SlHDA1) to assemble a chromatin remodeling complex. This complex then represses the expression of *SlWUS*, ultimately terminating the activity of FM cells [7]. *WUS* promotes the expression of CLAVATA3 (*CLV3*), which encodes a glycopeptide signaling peptide, and *CLV3*, in turn, suppresses *WUS* expression through arabinosylation mediated by *STP2*, which limits the number of stem cells and reduces fruit deformities [2,8]. The mechanisms of malformed tomato fruit in the CLAVATA-WUSCHEL pathway are well studied. Are there any new regulatory genes or pathways involved in tomato fruit malformation? This is still unclear.

It has been proven that poor pollen germination is linked to high levels of fruit malformation [9]. For pollen development and pollen tube elongation, pectinase is essential. It is a complex that includes three key enzymes: pectin lyase (PL), pectin methyl esterase (PME), and polygalacturonase (PG) [10,11,12,13,14]. The inhibition of *PL* genes resulted in irregular and shortened pollen tubes, with aberrant accumulations observed in the vicinity of the pollen germination grooves [15]. In tobacco, 30 PME genes were identified. *NtPME1* showed preferential expression in the stigma and ovary, and high expression levels in the pollen tube. Notably, silencing of *NtPME1* led to shrunken pollen grains and impaired pollen tubes [16]. Tang et al. [17] identified 81 PbrPMEs in pear, noting that inhibition of *PbrPME44*, *PbrPME59*, and *PbrPME11* disrupted the methyl esterification of pectin at the tips of pollen tubes, thereby hindering their growth. *OsFOR1*, a gene encoding a PG-inhibitory protein, was shown to elevate the count of stamens, carpels, paleas/lemmas, stigmas, and scales when suppressed, leading to deformity [18]. These studies indicated that pectinase genes are involved in fruit malformation by regulating pollen germination and development. Currently, no comprehensive bioinformatics analysis of pectinase genes in tomato has been conducted, and there are no reports proving that pectinase genes are associated with tomato fruit malformation.

In this study, we examined phenotypic characteristics and transcriptomic changes in wild type (WT) and malformed tomato fruit mutants, with a particular focus on novel candidate genes. Specifically, we analyzed the pectinase gene families associated with tomato fruit malformation. This research presents new insights into the regulation of fruit malformation, emphasizing the role of pectinase genes in mediating these defects.

## 2. Results

### 2.1. Morphological Identification of Deformed Fruit

The ‘QT2’ malformed-fruit mutant was distinguishable from the wild type (WT) ‘QT57’, whereas significant differences emerged during flower and fruit development. Compared with ‘QT57’, ‘QT2’ flowers developed an increased number of organs in all whorls, as well as a higher number of carpels—a phenomenon that ultimately resulted in anomalous fruits. Specifically, the fruits of QT-57 were round and regular in shape with small fruit navels, while those of ‘QT2’ exhibited an irregular shape, with cracks and protrusions on their surface (Figure 1A). The mean sepal and petal numbers in ‘QT57’ were both 5, whereas in ‘QT2’, the means were 9.00 and 9.25 at the original bud stage, respectively. At the post-bud stage, significant differences were observed in parameters such as bud longitudinal diameter, sepal number, petal number, stamen transverse diameter, ovary transverse diameter, and both transverse and longitudinal diameters of the pollen tube between the two varieties. At the bud stage, significant differences were also observed between the two varieties in the mean values of several traits. For ‘QT57’, the mean values of sepal and petal number were 5.25 and 5.75, respectively, while for ‘QT2’, they were 10 and 10.25, respectively. The transverse diameters of the bud, stamen, ovary, and pollen tube, along with the longitudinal diameters of the stamen, style, and pollen tube in ‘QT57’ were 3.10 mm, 2.72 mm, 7.77 mm, 1.81 mm, 6.19 mm, 0.52 mm, and 4.52 mm, respectively, while for ‘QT2’, they were 6.96 mm, 6.72 mm, 11.15 mm, 4.71 mm, 10.26 mm, 2.45 mm, and 6.86 mm, respectively. At the anthesis stage, the same differential indicators as those in the post-bud stage were identified in ‘QT2’ and ‘QT57’ (Figure 1B,C).

### 2.2. Transcriptome Analysis of Malformed Tomato Fruits

To explore the molecular responses associated with malformed tomato fruit, unopened flower buds (UP, more than 8 days before anthesis), half-opened flower buds (HP, approximately 4 days before anthesis), and fully opened flowers (FP) of ‘QT57’ and ‘QT2’ were subjected to RNA-Seq analysis. The total raw and clean reads for each sample ranged from 44.36 to 53.35 Mb and 42.01 to 49.76 Mb, respectively. The Q30 percentage ranged from 94.68% to 95.33%, and the GC content varied between 42.8% and 43.82% (Appendix A).

Differentially expressed genes (DEGs) were identified with *p*-value ≤ 0.05 and |log2FC| > 2. A total of 1133, 1115, and 2319 genes were upregulated, while 1512, 1732, and 3256 genes were downregulated in the three comparison groups: QT2_UP vs. QT57_UP, QT2_HP vs. QT57_HP, and QT2_FP vs. QT57_FP, respectively (Figure 2A,C). Among these, 819 genes were differentially expressed in all three comparisons, while 3294, 687, and 939 genes were specifically expressed in each comparison (Figure 2A).

Functional annotation of DEGs was performed for GO and KEGG pathways to evaluate functional enrichment. (Figure 2D,E). The top 30 and top 20 entries with PopHits ≥ 5 were analyzed for GO and KEGG enrichment during the ‘QT2’ and ‘QT57’ genotype in the UP, HP, and FP periods, respectively. Under the biological process category, the DEGs showed enrichment in pollen ectoderm formation, sporopollen biosynthesis, regulation of pollen tube growth, and pectinolytic processes. In the cellular component category, most DEGs were enriched in the cell wall, cytoplasm, plasma membrane, and pollen tubes. Among these processes and components, pollen germination and pollen tube growth are closely associated with fruit malformation, while pectinase genes play a critical role in pollen development and pollen tube elongation.

KEGG enrichment analysis revealed that the DEGs are predominantly involved in phytohormone signaling pathways, pentose and glucuronide interconversion, and the mitogen-activated protein kinase (MAPK) signaling pathway. Specifically, phytohormones such as abscisic acid (ABA) and gibberellin (GA) regulate callus accumulation and the intercellular movement of WUS, and further modulate floral meristem development—thereby influencing fruit malformation. Correspondingly, transcriptomic data analysis revealed that the three abovementioned KEGG-enriched pathways contained 96, 53, and 28 DEGs, respectively. Subsequent screening identified a total of 9, 19, and 3 genes that are differentially expressed in all UP, HP, and FP stages, respectively. These genes corresponded to the specific leading-edge genes within the three significant pathways. For instance, *EREB* was associated with the phytohormone signaling pathway, *LAT59* (pectate lyase P59) was identified in pentose and glucuronide interconversion, and *MPK3* was identified in the MAPK signaling pathway.

### 2.3. Expression Profiling of Differentially Expressed Genes

The top 23 differential genes in ‘QT2’ and ‘QT57’ during the UP, HP, and FP periods were clustered, revealing 12 down-regulated genes and 11 up-regulated genes (Figure 2B). Then, the 23 highly expressed DEGs at the UP, HP, and FP stages in ‘QT2’ and ‘QT57’ were compared with the DEGs from the KEGG enrichment analysis during the same stages. By comparing gene function annotations, we found that both the 23 highly expressed differential genes and the genes identified in KEGG enrichment analysis were involved in the same pathways, especially plant hormones, pectinases, mitosis, elongation factors, and FAS protein synthesis. These pathways may be associated with the occurrence of tomato fruit malformations. Based on this finding, 16 candidate genes associated with the above pathways were further screened out (Figure 2B).

The expression of 16 candidate genes was analyzed using RNA-seq and qRT-PCR (Figure 3C). RNA-seq results were consistent with qRT-PCR findings (Figure 3A,B). *MPK3* and *LOC101268544* were significantly more highly expressed in ‘QT2’ than in ‘QT57’ during the HP and FP periods; by contrast, *MPK3* expression in ‘QT2’ was significantly lower than that in ‘QT57’ during the UP period. Specifically, during the HP and FP periods, *MPK3* expression in ‘QT2’ was upregulated by 10.69- and 1.43-fold, respectively, relative to ‘QT57’, while *LOC101268544* expression in ‘QT2’ was upregulated by 6.66- and 8.72-fold, respectively, compared to ‘QT57’. Additionally, *LOC101257474* showed significantly higher expression in ‘QT2’ than in ‘QT57’ across all three periods (UP, HP, and FP), with respective upregulation fold changes of 4.22, 9.77, and 24.05 relative to ‘QT57’. The expression levels of *LOC101249702* and *EREB* in ‘QT2’ were significantly higher than those in ‘QT57’ during the UP and FP periods. In contrast, the remaining 11 genes exhibited significantly lower expression in ‘QT2’ compared to ‘QT57’ across all three periods.

Nine genes associated with tomato malformed fruit (Appendix A) have been reported in the studies by Castañeda et al. [7] and Wu et al. [2]. To verify whether these genes were involved in the formation of malformed fruit in ‘QT2’, their expression levels were analyzed (Figure 3A,B). The expression of *SlCRCa* was upregulated by 10.94-fold and 84.55-fold during the HP and FP periods, respectively, in ‘QT2’ compared to ‘QT57’. *SlCRCb* showed significant differences between ‘QT2’ and ‘QT57’ during the UP and FP periods, with expression levels of 35.96 and 34.1 in ‘QT2’ compared to 1 in ‘QT57’. *SlWUS* and *SlTPL* were significantly expressed only during the HP period, while *SlIMA* exhibited notable expression across the UP, HP, and FP periods in both ‘QT2’ and ‘QT57’. *SlKNU*, *SlTAG1*, and *SlTAGL1* were prominently expressed during the UP and FP periods, whereas *SlHDA1* showed significant expression during the HP and FP periods. Additionally, the expression of *SlCLV3* was upregulated by 50-fold during the UP period in the ‘QT2’.

### 2.4. Pectinase Genes Involved in Malformed Tomato Fruit Formation

Six pectinase genes were identified from the 16 candidate genes screened: one PL gene (*LAT56*), three PME genes (*LOC101260471*, *LOC101244677*, *LOC101267809*, *LAT59*), and one PG gene (*LOC101263150*). These genes exhibited significant differences in expression between ‘QT2’ and ‘QT57’ during the UP, HP, and FP periods. According to previous studies [19,20,21,22], the pectinase gene family comprises a total of 34 genes. Besides the six pectinase genes identified in this study, there were an additional 28 genes, which are specifically expressed in flowers (Appendix A). To explore their roles further, we analyzed the expression profiles of these 28 genes using transcriptome data. Heat map analysis revealed that all 28 genes displayed differential expression. Of these, *SlPG24-9*, *SlPG24-1*, *SlPG24-8*, *SlPG15*, *SlPG24-4*, *SlPG17-2*, and *SlPG24-6* were highly expressed in aberrant flowers, while the remaining genes were predominantly expressed in normal flowers (Figure 4A). qRT-PCR results indicated that, among these 28 genes, 19 showed significant differential expression between ‘QT2’ and ‘QT57’ during the UP, HP, and FP periods. Additionally, eight genes exhibited significant expression in two periods, and one gene showed significant differential expression in a single period. These findings indicate that pectinase genes may contribute to regulating malformed tomato fruit development (Figure 4B).

### 2.5. Identification of the Pectinase Gene Family

The characterization of pectinase genes was helpful for understanding their role in regulating malformed fruit development. Previous studies conducted a comprehensive bioinformatics analysis on the PG genes within the pectinase gene family. However, only preliminary analyses such as sequence alignment were performed on the PL and PME genes [20,21,22]. Hence, this study performed an in-depth bioinformatics analysis on pectinase genes. A total of 34 pectinase genes, comprising six identified in the present study and 28 from previous research, were subjected to bioinformatics analysis. These 34 genes included 6 PLs, 15 PGs, and 13 PMEs. Detailed information on the 34 genes was provided in Appendix A.

#### 2.5.1. Chromosomal Mapping

To reveal the distribution patterns and evolutionary relationships of pectinase genes, chromosomal localization was conducted. The results revealed that *PL* genes were distributed on chromosomes 2, 3, and 5; *PME* genes on chromosomes 1, 3, 5, 6, 7, and 12; and *PG* genes on chromosomes 2, 3, 4, 6, 7, 10, and 12. No *pectinase* genes were found on chromosomes 8, 9, and 11 (Figure 5A).

#### 2.5.2. Analysis of Protein Physicochemical Properties and Structure

To infer the fundamental molecular properties and potential functional characteristics of pectinase proteins, protein-related analyses were conducted. Analysis of protein physicochemical properties showed that PL genes had 398–616 amino acids, 44,277.08–68,766.17 Da molecular weights, and 6.92–8.68 isoelectric points. PME genes had 102–652 amino acids, molecular weights of 11,505.02–69,716.4 Da, and isoelectric points of 5.22–9.32. PG genes had 246–467 amino acids, molecular weights of 27,096.2–50,954.28 Da, and isoelectric points of 5.9–9.19 (Appendix A). Secondary structure prediction revealed that the α-helix, extended strand, β-turn, and irregular curl were found in PL, PME, and PG proteins (Appendix A). In terms of tertiary structure, the α-helix and β-folding counts for PL genes ranged from 8 to 14 and 31–51, respectively; for PME genes, 5–22 and 8–37, respectively; and for PG genes, 3–9 and 23–43, respectively (Figure 5C).

#### 2.5.3. Analysis of Subcellular Localization

The function of proteins is closely associated with the subcellular compartments in which they are located; therefore, we performed a subcellular localization analysis of pectinase genes. Subcellular localization analysis predicted that *PL* and *PME* genes were predominantly distributed in the extracellular matrix; notably, *SlPL7* was predicted to be in the nucleus. *PG* genes were primarily localized to the nucleus, with *SlPME9* specifically predicted in the endoplasmic reticulum (Appendix A). Therefore, further experiments were conducted using *SlPL7* and *SlPME9* for verification. The results revealed that *SlPL7* was localized in the nucleus, while *SlPME9* was detected in both the cell membrane and endoplasmic reticulum—results that are highly consistent with the predictions (Figure 5B).

#### 2.5.4. Evolutionary Analyses

The evolutionary analysis of pectinase genes helps to clarify the structural variations in gene families that occur during evolution and their association with functional differentiation. The evolutionary tree was constructed following multiple sequence alignment (Figure 5D). The analysis revealed that the tree categorized 33 genes into four classes, comprising 10, 3, 2, and 18 genes, respectively. Among these, PG was grouped into three taxa: I, II, and III, while PL and PME were classified under taxon IV. This classification suggested a close relationship between PL and PME, potentially indicating similar functions.

#### 2.5.5. Gene Structure and Promoter Analyses

Gene structure and promoter analysis help to predict the expression regulation patterns and potential physiological functions of genes. Gene structure analysis revealed that PL genes contained 3 to 13 exons and 2 to 12 introns, *PME* genes featured 1 to 8 exons and 0 to 7 introns, while PG genes exhibited 4 to 9 exons and 3 to 8 introns. Twelve motifs were identified via MEME. PL genes contain Pec_lyase_N and Amb_all domains. PME genes included the PMEI and Pectinesterase domains. PG genes were characterized by the PbH1 domain. The cis-acting promoter elements of pectinase genes were categorized into four groups. The first group was associated with hormone responses. The second group included elements linked to meristem organization, seed-specific regulation, chloroplast mesophyll cell differentiation, and endosperm expression. The third group comprised elements responsive to abiotic stresses. The fourth category encompassed elements involved in light responsiveness. (Figure 5E).

#### 2.5.6. Protein Network Interaction Analysis

The analysis of how pectinase genes interact with each other revealed how they work together to regulate the development of malformed fruit. The protein interaction analysis (Figure 5F) revealed 98 pairs of protein interactions between PL and PME. SlPME30, SlPME39, SlPME78, SlPG38-1, SlPG44, SlPG24-8, SlPG56-2, SlPG56-3, SlPG45, SlPG17-2, SlPG8, SlPG38-2, and SlPG36, none of which were involved in interactions with other proteins. These enzymes were primarily enriched in three key biological processes: the pectin catabolic process, carbohydrate metabolic process, and pentose glucuronide interconversion pathway.

## 3. Discussion

Fruit malformation poses a significant constraint on global fruit production, leading to substantial economic losses. While the molecular mechanisms behind malformed tomato fruits have been linked to the CLAVATA-WUSCHEL pathway, the involvement of other regulatory pathways remains largely unexplored. In this study, we conducted a phenotypic comparison between the wild type ‘QT57’ and the malformed fruit strain ‘QT2’. Notable differences in floral organization were observed, with malformed flowers exhibiting traits such as larger buds, an increased number of sepals and petals, and thick, irregular styles compared to normal flowers. These differences became increasingly pronounced as floral organs developed, indicating that specific genes may regulate the formation of deformed flowers, ultimately resulting in malformed fruits.

### 3.1. The CLAVATA-WUSCHEL Pathway’s Role in the Formation of ‘QT2’ Fruit Deformities

Some genes mediating the malformed fruit by regulating carpel development have been identified. Tomato *SlCRCa* and *SlCRCb* positively regulate floral meristems through a compensatory and partially redundant mechanism. Additionally, *SlCRCa*, *SlCRCb*, *SlIMA*, and *SlKNU* interact with *SlHDA1* and *SlTPL1* to assemble a chromatin remodeling complex. This complex then represses the expression of SlWUS, ultimately terminating the activity of FM cells. This process actively ensures the normal formation of flowers and fruits [4,5,6]. WUS promotes the expression of CLV3, and CLV3, in turn, suppresses WUS expression through arabinosylation mediated by STP2, which limits the number of stem cells and reduces fruit deformities [2,8]. To investigate whether the formation of ‘QT2’ deformed fruit is regulated by the CLAVATA-WUSCHEL pathway genes, gene expression was analyzed in ‘QT57’ and ‘QT2’ during the UP, HP, and FP periods. qRT-PCR and transcriptome sequencing analyses showed that *SlCRCa*, *SlCRCb*, and *CLV3* are barely expressed in the WT ‘QT57’. In contrast, their expression levels are significantly upregulated in the malformed-fruit mutant ‘QT2’. Specifically, *SlCRCa* was highly expressed in ‘QT2’ during the FP stage, while *SlCRCb* exhibited high expression during the UP and HP stages. This expression pattern suggests that *SlCRCa* and *SlCRCb* may regulate floral organ development in ‘QT2’ through a complementary mechanism, which was consistent with the findings of previous studies [7]. Mechanistically, the high expression of *SlCRCa* and *SlCRCb* in ‘QT2’ likely prevents the inhibition of stem cell activity in FM after floral organ formation. This delays FM termination, ultimately resulting in malformed fruits characterized by additional carpels and extra locules. Additionally, the elevated *CLV3* expression in ‘QT2’ may disrupt the normal restriction of *WUS* expression. This disruption increases floral organ number, further contributing to the formation of malformed fruits. Subsequently, the roles of *CLAVATA-WUSCHEL* pathway genes and key pectinase genes in the formation of malformed fruits in ‘QT2’ will be verified through overexpression and gene editing. Additionally, their regulatory mechanisms on tomato malformed fruits will be elucidated via gene function complementation assays and protein interaction analysis.

### 3.2. RNA Sequencing Reveals Six Pectinase Genes Linked to Tomato Fruit Malformation

‘QT57’ and ‘QT2’ tomato materials during UP, HP, and FP periods were subjected to RNA-Seq analysis to uncover new regulatory pathways related to malformed tomato fruit. RNA-Seq analysis identified 1133, 1115, and 2319 up-regulated differential genes, along with 1512, 1732, and 3256 down-regulated differential genes during the UP, HP, and FP periods, respectively. These differential genes may play a pivotal role in the development of deformed fruit. Sixteen key candidate genes were further identified and analyzed using qRT-PCR, revealing significant expression differences between ‘QT57’ and ‘QT2’, which implies their potential involvement in the formation of ‘QT2’ deformed fruit. Among these candidate genes, six pectinase-related genes were identified: *LAT56* (*PL* gene), *LOC101260471*, *LOC101244677*, *LOC101267809*, *LAT59* (*PME* gene), and *LOC101263150* (*PG* gene) [13]. These genes showed lower expression in ‘QT2’ compared to ‘QT57’, suggesting their negative regulatory role in fruit malformation. Additionally, these six pectinase genes demonstrated specific high expression in floral tissues [19,20,21,22], corroborating the findings of Jang et al. [18] and Li et al. [10], who confirmed that PG genes influence fruit malformation through overexpression and silencing analyses.

### 3.3. Pectinase Genes May Regulate Tomato Fruit Malformation Through the Gibberellin-WUS Pathway, Carbohydrate Metabolism, or Cell Wall Metabolic Disorder

Pectinase is a complex of enzymes, including PL, PME, and PG, whose activity is closely associated with fruit firmness [23,24]. However, the functions of pectinase genes and their molecular mechanisms in deformed fruit remain poorly understood. In this study, six pectinase genes significantly downregulated in the malformed fruit mutant ‘QT2’ were identified as key candidate genes for fruit deformation through RNA-seq, qRT-PCR, and bioinformatics analysis. Additionally, a gene family analysis of these pectinase genes in tomatoes was conducted. Promoter analysis revealed that all *pectinase* genes—except *SlPL2*, *SlPL11*, and *SlPG56-2*—possess gibberellin-responsive elements, suggesting that gibberellin may influence pectinase gene expression and the development of malformed fruits. Wu et al. [25] demonstrated that elevated gibberellin levels promote *WUS* gene expression, increasing the number of locules and contributing to fruit malformation. Furthermore, short-term cold stress raised ABA levels, enhancing callus accumulation, while the application of exogenous GA reduced callus formation, restored *WUS* intercellular movement, and facilitated normal FM development, thereby reducing fruit malformation.

Protein interaction network analysis revealed extensive interactions among pectinase enzymes. These enzymes were primarily enriched in three key biological processes: the pectin catabolic process, carbohydrate metabolic process, and pentose glucuronide interconversion pathway. León-Burgos et al. [26] reported that under high fruit load, coffee plants exhibit a significant decrease in total soluble sugar content, accompanied by an increase in the number of malformed fruits. In a separate study, Lu et al. [27] combined physiological and transcriptomic analyses and found that carbohydrate reserves were significantly reduced in malformed coconut fruits. These results indicate that pectinase genes may regulate the formation of malformed fruits by influencing carbohydrate metabolism. Notably, in malformed coconut fruits, the activities of cell wall remodeling enzymes—including cellulase (CEL), polygalacturonase (PG), and pectinesterase (PE)—were significantly enhanced [27]. This result was consistent with our study’s identification of key pectinase genes that regulate tomato fruit malformation. As important cell wall hydrolases, pectinases play a critical role in maintaining normal cell wall structure. Abnormal pectinase activity during fruit development leads to extensive cell wall degradation; the loosened cell walls then lose the ability to support the normal fruit shape, thereby making fruits prone to deformities such as cracking and indentation [28,29].

## 4. Materials and Methods

### 4.1. Plant Material and Growth Conditions

The ‘QT2’ malformed tomato fruit mutant was generated from natural variation. During the cultivation of the ‘QT2’ tomato variety in the Qinghai–Tibetan Plateau region of China (with an average altitude of 3500–4000 m, an annual average temperature of 0–6 °C, an annual precipitation of 10–700 mm, a relative humidity of 30–60% and an annual sunshine duration of 2500–3600 h), a single plant bearing only malformed fruits was found in the field when the variety was propagated to the F_5_ generation. After this plant was preserved for seeds and propagated separately, all the fruit produced by its progeny showed the malformed phenotype. The progeny was then subjected to continuous self-pollination for three generations to achieve genetic purity, and a genetically stable ‘QT2’ malformed-fruit mutant was finally obtained. The tomato genotype ‘QT57’ served as the wild type (WT). The tomato materials were preserved in the Key Laboratory of Vegetable Genetics and Physiology at Qinghai University. The plants were cultivated at the Horticultural Innovation Base of Qinghai University under standard field management practices.

### 4.2. Phenotypic Characterization of Tomato Flowers and Fruits

Phenotypic characterizations were assessed using at least six flowers from ‘QT57’ and ‘QT2’ during the anthesis stage. Ten morphological characteristics in ‘QT2’ and ‘QT57’ were analyzed during four developmental stages: the original bud stage (buds becoming evident, approximately 4 mm in length), the post-bud stage (buds more distinct but with sepals not yet separated), the bud stage (sepals slightly separated, revealing petals and stamens), and the anthesis stage (petals fully expanded and perpendicular to the style). These traits included the transverse and longitudinal diameters of the bud, stamen, and pollen tube; the transverse diameter of the ovary; the transverse and longitudinal dimensions of the style; and the number of sepals and petals. The collected data were averaged for analysis. Optical microscopy studies were conducted following the methodology outlined in Lozano et al. [30].

### 4.3. RNA-Seq Analysis

Based on the morphological identification of deformed tomato fruits in ‘QT2’ and ‘QT57,’ unopened flower buds (earlier than 8 days before anthesis), half-unopened flower buds (earlier than 4 days before anthesis), and fully-opened flowers were selected for expression analysis. Each sample included three biological replicates, with five mixed samples taken per replicate. In total, 18 samples were used for RNA-seq analysis, with three replicates per group.

Total RNA was isolated using the Plant RNA Purification Reagent (Invitrogen, Carlsbad, CA, USA), and a NanoDrop 2000 spectrophotometer (Thermo Scientific, Rochester, New York, USA) was used to assess its purity. The RNA sequence was conducted on the Illumina NovaSeq 6000 platform (Ouyi Biological Company, Shanghai, China). Raw data were processed with Fastp software v0.23.4 [31], and the clean reads were then mapped to the tomato reference genome (ITAG3.1.0 SL3.1) via HISAT2 (v2.0.4, http://www.ccb.jhu.edu/software/hisat/index.shtml, accessed on 3 April 2024). Differential expression analysis was conducted with DESeq2 (v1.4.5, www.bioconductor.org/packages/release/bioc/html/DESeq2.html, accessed on 3 April 2024), applying thresholds of *p*-value ≤ 0.05 and |log2FC| > 2. Differentially expressed genes (DEGs) were further analyzed for functional enrichment using the GO and KEGG databases, leveraging the hypergeometric distribution algorithm.

### 4.4. qRT-PCR Analyses

Total RNA was extracted with the TaKaRa MiniBEST Plant RNA Extraction Kit (TaKaRa, Shanghai, China). RNA quality and purity were assessed with a NanoDrop 2000 spectrophotometer (Thermo Scientific, Rochester, NY, USA). qRT-PCR analysis was conducted on an Eppendorf real-time PCR machine (Hamburg, Germany) using the All-In-One 5X RT MasterMix and BlasTaqR 2X qPCR MasterMix (ABI, Shanghai, China), following the manufacturer’s protocol. Relative gene expression was calculated via the 2^−ΔΔCt^ method [32]. The CDS sequences used for qRT-PCR analysis are provided in Appendix A.

### 4.5. Gene Family Analysis

The gene IDs of pectinase genes were sourced from studies conducted by Yang et al. [20], Jeong et al. [21], and Ke et al. [22]. The Ensembl Plants database (https://plants.ensembl.org/index.html, accessed on 4 April 2024) was used to obtain their protein sequences, CDS sequences and GFF3 annotation files. Chromosomal localization was performed using TBtools (v2.089)**.** Conserved motifs of Pectinase genes were identified through the MEME suite (http://meme.nbcr.net, accessed on 4 April 2024). Protein structural domains were predicted using the SMART platform (https://smart.embl.de/, accessed on 5 April 2024). 2 kb upstream nucleotide sequences of the pectinase genes were submitted to the Plant CARE online tool (http://bioinformatics.psb.ugent.be/webtools/plantcare/html, accessed on 5 April 2024) via the “search for care” function to identify promoters. TBtools was then employed to visualize gene structures, protein structural domains, and cis-acting elements. Multiple sequence alignments were performed using the ClustalW program with default parameters, followed by phylogenetic tree construction with Molecular Evolutionary Genetics Analysis (MEGA) software v6.0, which was further optimized using iTOL (https://itol.embl.de/, accessed on 8 April 2024). Physicochemical properties of proteins were conducted on the ExPASy platform (https://www.expasy.org/, accessed on 8 April 2024). Subcellular localization predictions were performed using WoLF PSORT (https://wolfpsort.hgc.jp, accessed on 8 April 2024). Protein structures were predicted using SWISS-MODEL (https://swissmodel.expasy.org/, accessed on 8 April 2024) and SOPMA (https://npsa.lyon.inserm.fr/cgi-bin/npsa_automat.pl?page=/NPSA/npsa_sopma.html, accessed on 8 April 2024), respectively. Protein network interactions were analyzed via STRING (https://cn.string-db.org/, accessed on 8 April 2024) with a combined score threshold of ≥0.4.

### 4.6. Subcellular Localization

Subcellular localizations were analyzed by fusing each protein to green fluorescent protein (GFP). The CDS of *SlPL7* and *SlPME9* genes, excluding the stop codon, were amplified from cDNA using gene-specific primers (listed in Appendix A). The pCAMBIA2300-GFP vector was double-digested with restriction enzymes *Sac* I and *Xba* I at 37 °C for 3 h. Recombinant plasmids were constructed by homologous recombination using the ClonExpress II One Step Cloning Kit (Vazyme, Nanjing, China) following the manufacturer’s protocol. The verified recombinant plasmids were transformed into *Agrobacterium tumefaciens GV3101* strain and infiltrated into the leaves of 2- to 3-week-old Nicotiana benthamiana plants, following the protocol described by Robertson [33]. Infiltrated plants were maintained under long-day conditions (16 h light/8 h dark) at 22 °C for two days. Fluorescent signals were visualized using a Zeiss Observer Z1 fluorescence microscope (FV10-ASW, Teltow, Germany). Images were captured using a Zeiss AxioCam MRm camera and processed with Zeiss ZEN 2.3 software.

### 4.7. Data Analysis

SPSS 16.0 (SPSS Inc., Chicago, IL, USA) was used for data analysis. ANOVA with Tukey’s post hoc test was performed, and significance was set at *p* ≤ 0.05. The bar graph was drawn using Origin 9.0 (OriginLab, Northampton, MA, USA).

## 5. Conclusions

In conclusion, this study identified a novel regulatory pathway for tomato fruit malformation, whereby pectinase genes may exert their effects via the gibberellin-WUS pathway, carbohydrate metabolism, or cell wall metabolism. This pathway provides new target genes for the precise regulation of fruit malformation and offers significant reference value for practical production. Specifically, the incidence of malformed fruit can be reduced by: (1) precisely controlling the application of gibberellin-based regulators, (2) ensuring a balanced nutrient supply to maintain stable carbohydrate transport to the fruit, and (3) minimizing environmental stress on cell wall metabolism to avoid malformations related to abnormal cell wall synthesis.

## Figures and Tables

**Figure 1 ijms-26-10739-f001:**
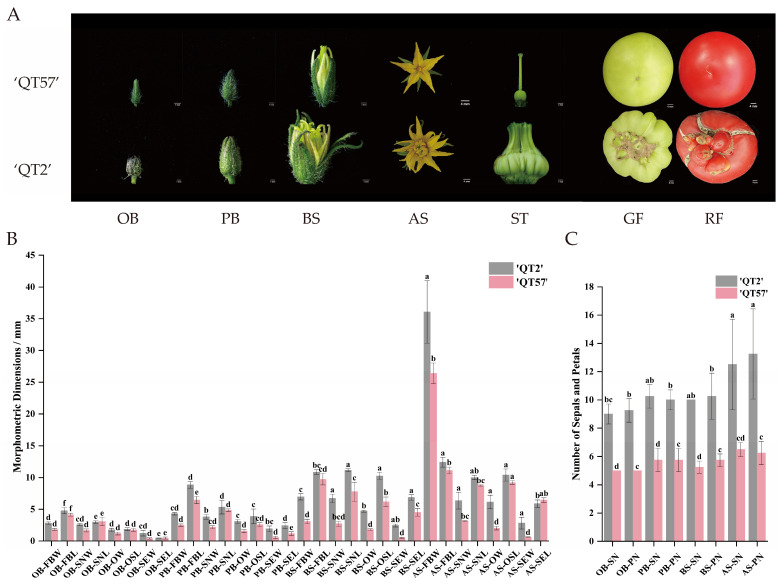
Flower buds and fruits in ‘QT57’ and ‘QT2’. (**A**) Bud and fruit phenotypes at different stages. (**B**) The number of sepals and petals in ‘QT57’ and ‘QT2’. (**C**) The characteristics of flower buds in ‘QT57’ and ‘QT2’. Where OB—Original bud stage; PB—Post-bud stage; BS—Bud stage; AS—Anthesis stage; FBW—Flower bud transverse diameter (mm); FBL—Flower bud longitudinal meridian (mm); SN—Sepal number (no.); PN—Petal number (no.); ST—Stigma; GF—Green-ripe fruit; RF—Red-ripe fruit; SNW—Stamen transverse diameter (mm); SNL—Stamen longitudinal meridian (mm).); SNW—Stamen transverse diameter (mm); SNL—Stamen longitudinal meridian (mm); OW—Ovary transverse diameter (mm); OSL—Ostyle longitudinal meridian (mm); SEW—Pollen tube transverse diameter (mm); SEL—Pollen tube longitudinal meridian (mm). SPSS 16.0 (SPSS Inc., Chicago, IL, USA) was used for data analysis. ANOVA with Tukey’s post hoc test was performed, and significance was set at *p* ≤ 0.05. Data are presented as the mean ± SE from three biological repeats. Different letters above the bars in the bar chart indicate statistical significance of differences.

**Figure 2 ijms-26-10739-f002:**
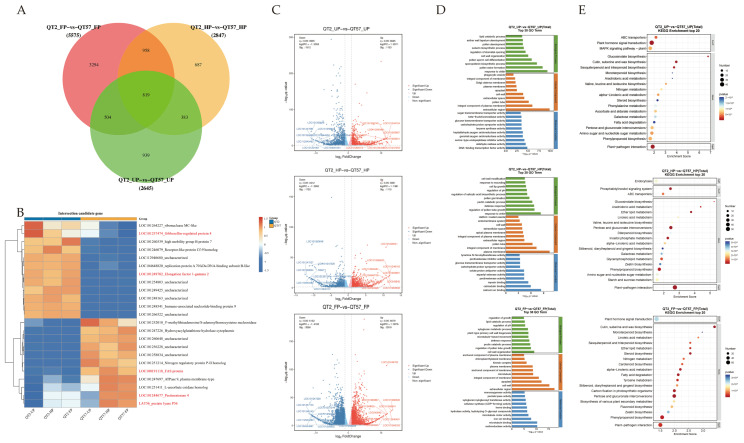
Differential gene screening. (**A**) differential gene Venn diagram plots. (**B**) Heatmap analysis of the top 23 differential genes in the transcriptome. The blue, white, and red color lumps indicated the low, medium, and high levels of expression, respectively; genes marked in red font represent candidate genes associated with KEGG pathways of plant hormones, pectinases, mitosis, elongation factors, and FAS proteins. (**C**) Differential gene volcano plots. (**D**) GO functional annotation. (**E**) KEGG functional annotation.

**Figure 3 ijms-26-10739-f003:**
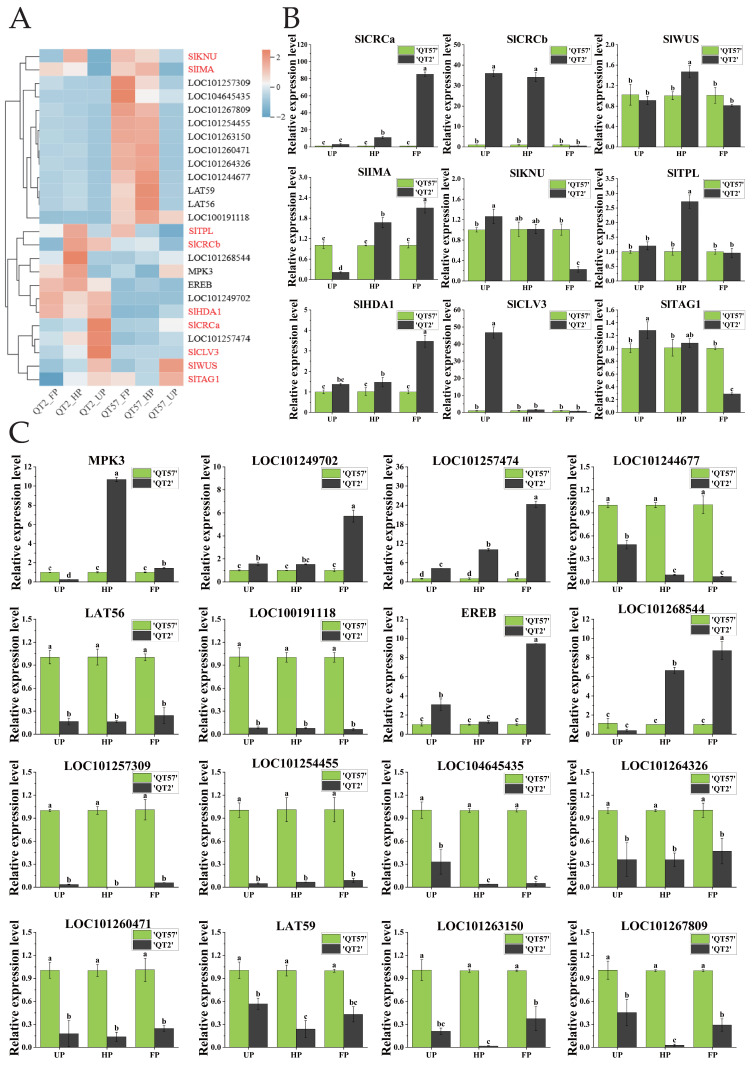
Analysis of candidate gene expression. (**A**) Transcriptome heatmap of the 16 candidate genes and the nine genes involved in the CLAVATA-WUSCHEL pathway; the genes in red are those involved in the CLAVATA-WUSCHEL pathway, and the genes in black are differentially expressed genes identified through transcriptome sequencing. The blue, white, and red color lumps indicated the low, medium, and high levels of expression, respectively; (**B**) RT-qPCR analysis of the nine genes involved in the CLAVATA-WUSCHEL pathway; (**C**) RT-qPCR analysis of the 16 candidate genes. The gray column represents ‘QT2’, and the green column represents ‘QT57’. SPSS 16.0 (SPSS Inc., Chicago, IL, USA) was used for data analysis. ANOVA with Tukey’s post hoc test was performed, and significance was set at *p* ≤ 0.05. Data are presented as the mean ± SE from three biological repeats. Different letters above the bars in the bar chart indicate statistical significance of differences.

**Figure 4 ijms-26-10739-f004:**
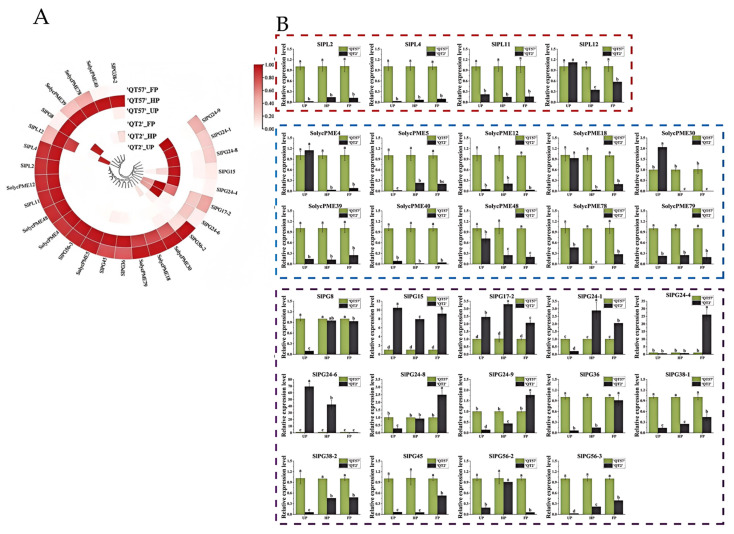
*Pectinase* gene expression analysis. (**A**) Circular clustering heatmap of the *pectinase* genes transcriptome. From light to dark red indicates an increase in gene expression levels; (**B**) RT-qPCR analysis of the *pectinase* genes. The red box indicates pectinolytic enzyme, the blue box pectin esterase, and the purple box polygalacturonase. The gray column represents ‘QT2’, and the green column represents ‘QT57’. SPSS 16.0 (SPSS Inc., Chicago, IL, USA) was used for data analysis. ANOVA with Tukey’s post hoc test was performed, and significance was set at *p* ≤ 0.05. Data are presented as the mean ± SE from three biological repeats. Different letters above the bars in the bar chart indicate statistical significance of differences.

**Figure 5 ijms-26-10739-f005:**
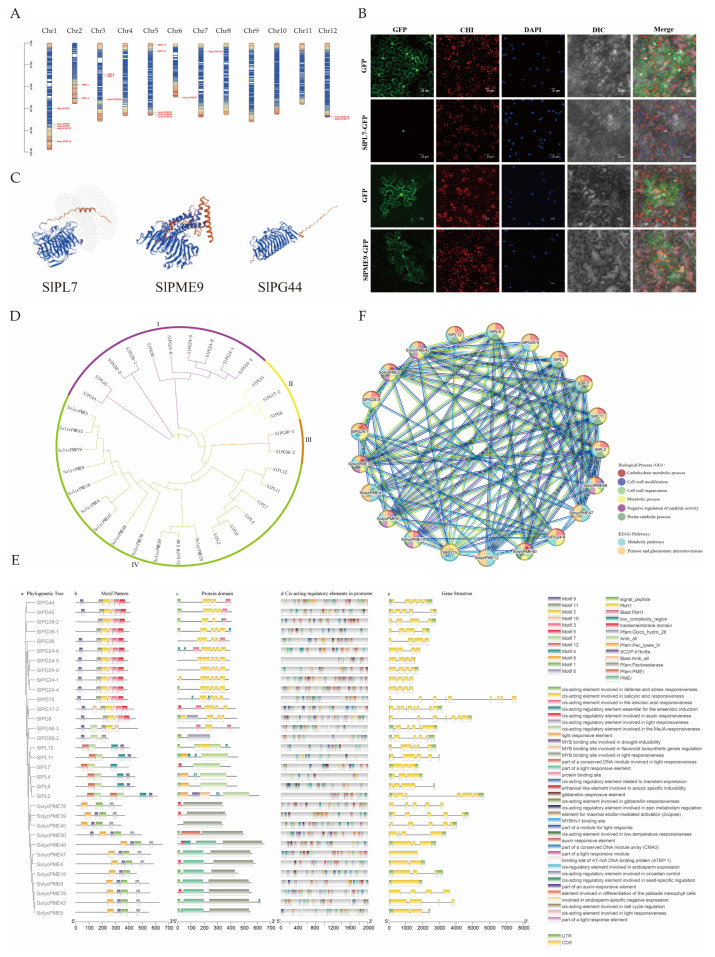
Bioinformatics analysis of pectinase gene family. (**A**) chromosomal localization; (**B**) subcellular localization of GFP, SlPL7-GFP, and SlPME9-GFP in the tobacco. The bars indicate 20 μm. GFP—green fluorescence field, CHI—chloroplast field, DAPI—nucleus field, DIC—bright field, Merge—merged field. (**C**) partial pectinase tertiary structure prediction, and all pectinase tertiary structures are shown in Appendix A; (**D**) phylogenetic tree; (**E**) prediction of gene structure, protein structural domains, and cis-acting elements; (**F**) protein network interactions.

## Data Availability

The RNA sequence data that support the findings of this study were submitted to the China National Genebank DataBase (CNGBdb) under accession numbers: CNP0007128.

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
