# Peer review of "Integrated Analysis of Transcriptomes and Pectinase Gene Families Reveals a Novel Pathway Mediating Tomato Fruit Malformation"

_ijms, 2025, doi:10.3390/ijms262110739_

Round 1

Reviewer 1 Report

Comments and Suggestions for Authors

The work of Wen et al. investigates the transcriptomic profile of tomato wild-type QT57 and a malformed-fruit mutant QT2 across three flower developmental stages. Given that tomato fruit malformation causes substantial yield losses in tomato production, understanding its molecular basis is of clear agronomic and economic interest. By performing phenotypical analyses, the authors confirm the QT2 mutant phenotype, supporting the relevance of performing RNAseq. The authors identify differentially expressed genes in WT/QT57 versus QT2 mutants at each developmental stages, and conduct gene ontology analyses to functionally characterize the deregulated genes. The study also includes a more focused analysis on specific candidate genes, including pectinase-related genes, which the authors propose as potentially relevant to the QT2 phenotype. The experimental design is good, with an adequate number of biological replicates and independent repetitions. However, the manuscript in its current form suffers from strong issues in data interpretation, lack of clarity regarding the rational of experiments and bioinformatic analyses. Furthermore, several figures are not readable and required more detailed captions. I encourage the authors to revise and resubmit, but in its current state, I do not consider that the manuscript is suitable for publication in International Journal of Molecular Sciences. Please see below for my major and minor points.

Major points

Please clarify the identification of the QT2 mutant. The method state that QT2 is “derived from natural variation”: are there any available genetic data or previous studies describing this line? If this is the first characterization of QT2, please state it clearly. This information should also be summarized in the Results section when introducing the mutant.

Figure1B: The plot is confusing as it combines variables with different units under the same y-axis. For example, some traits are expressed in mm (ie. BL-Flower bud longitudinal meridian), and others are counts (ie. N-Sepal number (no.)). I would suggest the authors to separate this plot into multiple panels according to their measurement units.

Line133: Please clearly indicate the p adjusted value and log2fc threshold chosen for significance, “criteria FDR 2” is not clear.

Line136-139: The Venn diagram does not show that the 819 genes are “co-expressed”. Indeed, it shows that there are 819 genes that are differentially expressed in all three comparisons (in QT2_UP vs. QT57_UP, QT2_HP vs. QT57_HP, and QT2_FP vs. QT57_FP). This is an important point as the diagram include both up- and down-regulated genes, thus the same gene could be up-regulated in one condition, and down-regulated in another, which is not co-expression. Furthermore, it is not correct to say that “3,294, 687, and 939 genes were exclusively expressed in QT2_UP vs. QT57_UP, QT2_HP vs. QT57_HP, and QT2_FP vs. QT57_FP”; indeed, these genes are specifically deregulated in each comparison, but these genes might still be expressed across all tissues and genotypes. Please rephrase the text and interpretation accordingly.

Line149-163: This section needs more than listing all significant GO and KEGG terms. The current description is broad and does not highlight the biological relevance of the findings. I would suggest the authors to focus on the significant terms relevant to their phenotypes (ie. mention terms related to pollen development and cell wall for example), and avoid detailing non-relevant terms such as the ones from the Cellular Component and Molecular Function categories.

Line161: Please specify where does these genes comes from? How they have been identified? Do these correspond to the specific leading-edge genes within the significant pathways?

Line167-169: It is not clear how the intersection between DEGs and KEGG pathways was performed. Please provide more detail (ie. Selection criteria? Overlap method used?)

Figure3A: This panel includes “the 16 candidate genes and the nine genes involved in CLAVATA-WUSCHEL pathway”; please use different color for each gene categories to improve figure readability. Furthermore, the representation here (dotplot) differs from Figure2B (heatmap), which make it very confusing for the readers. If possible, please use the same type of representation if the same type of data is displayed. Finally, the figure caption should also be more detailed (ie. what does dot size corresponds to?; color?).

Line172: MPK3 is not significantly higher in QT2 in UP (Figure3C). Please update the text.

Line179-180: The phrasing “exhibited significantly higher expression in 'QT57' compared to 'QT2'” is very confusing. Since QT57 is the wild type, results should be described relative to it. Same issue happened at Line342-343.

Line181: How these “Nine genes associated with the regulation of deformed tomato fruits”: have been identified? Please add more detail.

Line209: It is unclear where the 28 flower-specifically expressed genes come from. Were these identified in another study? Or newly detected in this dataset? Please clarify.

Figure4A: Please describe more precisely the plot in the caption. What does the color represent? And similarly, as for Figure3A and 2B, please use the exact same representation for homogeneity.  

Line227: The wording could be confusing for readers, as it could be understood that the current study identified six pectinase genes. However, the work primarily highlights pectinase genes showing different expression levels rather than discovering new ones. Please update the wording to avoid confusion.

Line230-291: This section feels disconnected from the rest of the manuscripts and lack biological contexts. In the current state it reads more like raw bioinformatics summary results. I would suggest the authors to clearly state the purpose of each analysis and why these are relevant to understanding the QT2 phenotype or the role of pectinase genes in fruit malformation? For example, what is the purpose of including chromosomal mapping? If some sub-sections are not directly relevant for the study, please consider removing this part.

Line323-325: The observed deregulation of SlCRCa and SlCRCb is not enough to confirm their direct involvement in fruit malformation; These changes could be indirect effects due to the deregulation of other upstream genes. I would suggest the authors to tone down this conclusion and mention that functional validation through genetic approaches (ie. KO, CRISPR…) is required to confirm their role.

Line333-335: A highest number of down-regulated genes does not support that the down-regulated genes play a more pivotal role that the up-regulated ones. There could be very relevant up-regulated genes too.

Minor points

Line16-20 and Line354-355: The terminology used to describe the method is confusing, and redundant. For example, “integrated transcriptomic data and expression profiling” is misleading as expression profile is a direct outcome of the transcriptomic analysis. Similarly, “via RNA-seq, qRT-PCR, and bioinformatics” is redundant as bioinformatics is intrinsic to RNAseq data analysis. Please rephrase for clarity.

Figure2A: Panel A appears to be a Venn diagram, not a “Wayne plot.” Please correct the terminology.

Figure2B: The caption is not clearly described: what does these genes corresponds to? What does the color scale represent, why are some genes colored in red?

Figure2C, D, and E: Please avoid using lower case letter (a), (b), and (c) it could be a bit confusing for the readers. Furthermore, the GO and KEGG plots are difficult to read, please increase font size.

Line162-163: I am not sure that this information adds much value? Indeed, stated that some genes have an absolute log2FC >2 is not informative per se. Please elaborate on the biological relevance if you wish to keep this sentence, or consider removing it.

Figure3B and C: Please increase font size, and make the QT57 sample appeared first in the barplot (as it is confusing for reader to display the QT2 mutant on the left).

Figure4B: Please increase font sizes.

Line402: What does “threshold of q2” corresponds to? Please specify log2fc and p adjusted value instead.

Typos

Line69: “idtneified”

Line151: “stages” (genotype?)

Reviewer 2 Report

Comments and Suggestions for Authors

A comprehensive analysis of pectinase transcriptomes and gene families was performed, revealing a pathway mediating tomato fruit malformations. Some comments:

A separate subsection on statistical analysis should be provided in the Materials and Methods section.

The fluorescence microscopy method should be described in more detail in the Materials and Methods section.

The y-axis in Figure 1 should be labeled.

It is desirable to determine the ploidy of deformed fruits. A more detailed description of how the molecular mechanisms involved in the formation of deformed fruits are regulated by CLAVATA-WUSCHEL feedback loops is needed.

A more detailed description of the mechanisms that cause reduced expression, in addition to nutrient deficiency in tomatoes and the effects of cold, is needed.

A separate Conclusion section should be provided.
